# Senescent Stromal Cells in the Tumor Microenvironment: Victims or Accomplices?

**DOI:** 10.3390/cancers15071927

**Published:** 2023-03-23

**Authors:** Minghan Ye, Xinyi Huang, Qianju Wu, Fei Liu

**Affiliations:** 1State Key Laboratory of Oral Diseases, National Clinical Research Center for Oral Diseases, National Center for Stomatology, West China School of Stomatology, Sichuan University, Chengdu 610065, China; 2Department of Orthodontics, School and Hospital of Stomatology, Cheeloo College of Medicine, Shandong University & Shandong Key Laboratory of Oral Tissue Regeneration & Shandong Engineering Laboratory for Dental Materials and Oral Tissue Regeneration, Jinan 250100, China; 3Stomatological Hospital of Xiamen Medical College, Xiamen Key Laboratory of Stomatological Disease Diagnosis and Treatment, Xiamen 361008, China; 4Shanghai Ninth People’s Hospital, School of Medicine, Shanghai Jiao Tong University, Shanghai 200011, China

**Keywords:** cellular senescence, stromal cell, tumor microenvironment, cancer-associated fibroblast, immune infiltration

## Abstract

**Simple Summary:**

Cellular senescence is a defensive response of cells to external stresses and occurs in a wide range of physiological and pathological processes, including neoplasms. Although tumor cell senescence may serve as a barrier to tumor proliferation, more studies have shown that stromal cell senescence in the tumor microenvironment contributes to the growth and regulation of tumor cells through senescence-associated secretory phenotypes. Herein, a review of the role of senescent stromal cells in the tumor microenvironment suggests that senescent stromal cells may be an accomplice in promoting tumor growth. This review contributes to understanding stromal cells in the tumor microenvironment and guides future research in the field.

**Abstract:**

Cellular senescence is a unique cellular state. Senescent cells enter a non-proliferative phase, and the cell cycle is arrested. However, senescence is essentially an active cellular phenotype, with senescent cells affecting themselves and neighboring cells via autocrine and paracrine patterns. A growing body of research suggests that the dysregulation of senescent stromal cells in the microenvironment is tightly associated with the development of a variety of complex cancers. The role of senescent stromal cells in impacting the cancer cell and tumor microenvironment has also attracted the attention of researchers. In this review, we summarize the generation of senescent stromal cells in the tumor microenvironment and their specific biological functions. By concluding the signaling pathways and regulatory mechanisms by which senescent stromal cells promote tumor progression, distant metastasis, immune infiltration, and therapy resistance, this paper suggests that senescent stromal cells may serve as potential targets for drug therapy, thus providing new clues for future related research.

## 1. Introduction

Cellular damage occurs as a result of environmental and intrinsic stimulation during proliferation and metabolism. Cells engage a variety of defense mechanisms in response to the phenomenon, including cellular senescence, to deal with tissue deterioration and prevent tumorigenesis. Cellular senescence, which refers to an irreversible state of cell cycle arrest, was first discovered by Hayflick and Moorhead [1]. After decades of research, cellular senescence has been studied intensively and is considered to be the response to many internal and external stimuli [2]. The transformation into senescent cells involves a variety of changes, including morphological alterations [3], macromolecular damage [4], cell cycle arrest [4], and metabolic changes [2,5]. However, as the understanding of cellular senescence continues to improve, some studies have found that although the cell cycle arrest caused by cellular senescence is usually irreversible, in some cases, especially in cancer cells, the re-entry of senescent cells into the cell cycle still occurs under certain conditions [6,7].

The link between cellular senescence and cancer has long been an attractive topic. For oncogene-induced senescence (OIS), p16/pRB and p53/p21 are crucial pathways for inducing cellular senescence [2,8,9], and anomalies in some of the key genes or substances involved in the two pathways are also considered biomarkers for detecting cellular senescence [10]. OIS was identified by several studies and was considered an important mechanism to suppress tumor growth [11,12,13,14]. Thus, cellular senescence is seen as a potential antitumor therapeutic target, and the “one-two punch” approach combined with pro-senescence and senolytic therapy holds promise for new approaches to cancer treatment [9,15,16]. Nevertheless, emerging evidence proved the Jekyll and Hyde effect of cellular senescence in tumor growth and development [17]. The dark side of cellular senescence promotes tumor progression, distant metastasis, immune infiltration, and resistance against antitumor therapy [18]. These negative effects are tightly correlated to senescence-associated secretion phenotype (SASP). SASP is defined as a collective term for pro-inflammatory cytokines, growth factors, and extracellular modifiers secreted by senescent cells [19,20]. The major SASP factors include interleukins, chemokines, growth factors, and so on, which exert the biological functions of senescent cells [2,21]. It is located downstream of the process of cellular senescence and acts as a stable functional arm, influencing neighboring cells through paracrine secretion to exert the double-edged sword effect of cellular senescence [9,20,21]. Research on SASP has deepened the understanding of the biological role of cellular senescence in cancer development, changing the previously held view of cellular senescence as an antitumor defense mechanism. Thus, the underlying pro-tumorigenesis effect of SASP is an urgent issue to be addressed.

Except for cellular senescence, the tumor microenvironment (TME) is also an indispensable component of tumorigenesis and metastasis. The concept of the TME, first proposed by Hanahan et al. in 2011, is a microenvironment constructed by multiple stromal cells, which contains parenchyma (i.e., tumor cells and tumor stem cells) as well as mesenchyme, complementing and enriching Paget’s theory of “seed and soil” as well as Virchow’s conjecture about the relationship between tumor and inflammation [22,23,24,25]. Its role is to recruit and adapt untransformed cells through multiple intercellular communication pathways after the malignant transformation of cells caused by oncogenic signals, ultimately leading to tumor formation [25]. Increasing evidence indicates that non-tumor cells in the TME are affected by SASP and undergo abnormalities. Coppé and colleagues found SASP factors target infiltrated immune cells and stromal cells in the TME to induce paracrine senescence, facilitating cancer cell epithelial–mesenchymal transitions (EMTs) and invasiveness [18,20]. However, the link between cellular senescence and the TME goes beyond this. Krtolica et al. discovered that the cellular senescence phenomenon occurs in mesenchymal cells, showing that tumor cells are not the only ones that senesce [26]. Subsequent research discussed in this review progressively revealed the criticality of the onset of cellular stromal senescence playing a role in the tumor.

Unfortunately, the understanding of the role of senescent stroma in the TME has been very limited so far. To date, several important questions have been asked:Which types of stromal cells undergo senescence in the TME and their corresponding biological changes?Which signals do tumor cells transmit to and receive from the senescent stromal cells?What is the role of infiltrated senescent immune cells in tumor development and progression?What is the role of senescent stromal cells to develop therapy resistance in the TME?

A recent review discussed the potential of exploiting senescence for the treatment of cancer [9]; however, how to circumvent the negative effects of SASP remains to be tackled. The heterogeneity of SASP effects, which are closely related to parameters such as tumorigenic location and tissue typing, also implicates prognosis. In another recent review, the current understanding of the onset and function of senescent cells in the TME is reviewed, and gaps and challenges in the field are highlighted [27]. This review focuses on analyzing and summarizing the role of different types of senescent non-tumor cells. However, there is still a gap in the work summarizing the heterogeneous nature of senescent stroma effects in this field. Therefore, in our review, we focus on the diverse nature of senescent stroma effects in the TME at this stage and summarize the prospects to provide ideas and references for subsequent research work in the field of cellular senescence and cancer.

## 2. The Discovery of Senescent Stroma

The senescence of tumor cells was recognized earlier. In contrast, stromal cell senescence has only gradually gained attention with the introduction of the TME concept and the maturation of research in this area. This section presents biomarkers of senescent cells and evidence of tumor stroma senescence, thus reviewing the course of research in this field.

### 2.1. Biomarkers of Senescent Cells

Cellular senescence is highly heterogeneous, with different cell types, tissues, organs, and triggers of senescence leading to differences in the degree, mechanisms, and effects of cellular senescence [2,9,15]. To effectively detect senescent cells for experimental and therapeutic purposes, it is crucial to find biomarkers with higher specificity. Generally, the hallmarks of cellular senescence include cell cycle withdrawal, macromolecular damage, deregulated metabolism, and secretory phenotype, but all of these hallmarks are diverse among different types of cellular senescence [2]. 

The biomarkers used at this stage are closely related to the hallmarks mentioned above. As cell cycle inhibitors, p16 and p21 are common biomarkers to detect cellular senescence [2,28,29]. In addition, senescence-associated β-gal (SA-β-gal), which reflects the upregulation of lysosome mass [2,30,31], and lipofuscin, which reflects the protein damage in senescent cells, are widely used as the markers of cellular senescence [30,32]. Among them, p16 is of the highest specificity [10]. However, certain non-senescent cells also share the cellular senescent features mentioned above [10,30]. Thus, it is recommended to access the senescence level using multi-marker approaches, which was attentively reviewed by a prior publication [2].

### 2.2. Evidence of the Occurrence of Cellular Senescence in Tumor Stroma

Decades after the discovery of cellular senescence, several studies have identified the phenomenon of tumor cell senescence, opening a whole new path for understanding the mechanism of tumorigenesis and developing new antitumor therapies. As research continued and the concept of the TME was introduced, the importance of the tumor stroma was gradually recognized.

In vitro and in vivo studies have provided new clues to the discovery of cellular senescence in stromal cells. Pazolli et al. treated mouse papilloma tissue with 7,12-dimethylbenz[a]anthracene/12-O-tetradecanoylphorbol-13-acetate (DMBA/TPA) to induce senescence, showing elevation of p16 and SA-β-gal in the stromal compartment prior to the epithelial compartment of papillomas [33]. Another study described a luciferase knockin mouse (p16^LUC^), a model faithfully reflecting the expression level of p16, observing p16 activation in the cells of peritumor stromal compartment [34]. A subsequent study using the same model yielded similar findings in Lewis lung cancer and spindle cell tumor xenografts [35]. As to which cell types are prone to senescence-related phenotype alterations, there are variations between studies. One study selected Nras^G12V^ to induce senescence in murine hepatocytes, in which senescent hepatocytes were surrounded by fibroblasts, lymphocytes, and plasma cells, most of which were characterized as p16- and p21-positive [36]. Additionally, the ratio of p21^+^ cells in tissues from human colon sessile serrated adenoma (SSA) was significantly higher versus tissues from the normal colonic crypt [36]. Meanwhile, senescent T cells, commonly characterized by CD27 and CD28 deficiency, were proven to undergo telomerase shortening and high expression of senescence-specific markers in response to the induction of cancer cells [37]. In another study, after OIS was triggered by oncogenic H-Ras in the old skin of mice, senescent cells were found in the stromal compartment and were rapidly cleared; this phenomenon led the researchers to speculate that these cells were infiltrating immune cells such as lymphocytes or T cells since senescent epidermal cells would persist longer [38,39]. In a study of an obesity-associated hepatocellular carcinoma (HCC) model in mice, senescent hepatic stellate cells induced by deoxycholic acid were reported [40]. In summary, different stimuli, species, systemic conditions, organs, and tissue types all lead to variations in the types of cells that experience cellular senescence (discussed in detail later). Nonetheless, it must be affirmed that cellular senescence of stromal cells in the TME is pervasive and is capable of interacting with tumor cells and thus may be involved in tumor initiation and progression. 

As the most specific senescence marker known to date, p16 reflects the distribution of senescent cells in tissues. Several studies have been conducted to detect the expression and distribution of p16 in different tumors to clarify the value of p16 as a biomarker in tumor early diagnosis and prognosis prediction, which provides evidence of stromal cell senescence in samples from human tumors. Two studies found that p16 was diffusely expressed in uterine serous carcinoma [41,42]. Another study included 35 patients with endometrial polyps, of which p16+ cells were observed in 31 cases, mainly in the areas around the glands where the characteristic fibrous stroma of the polyp persisted [43]. In a cohort including 124 cases of endometrial lesions, stromal p16 expression was significantly correlated with endometrial carcinomas rather than benign and preneoplastic lesions [44]. This trend of increasing stromal p16 expression levels accompanied by increasing malignancy was also noted in ovarian tumors [45]. For gastric-type mucinous carcinoma, a type of cervical malignant lesion characterized by aggressive behaviors, p16 was significantly expressed in stromal cells. On the contrary, p16 was not or was weakly expressed in cancer cells [46]. Furthermore, stromal p16 overexpression was observed in atypical polypoid adenomyoma [47], breast carcinoma [48,49], and adult granulosa cell tumors of the ovary [50], suggesting the occurrence of stromal cell senescence. In one other study, senescent T-cell infiltration was identified in the stromal compartment of colorectal cancer tissues [51]. Overall, there is abundant evidence that stromal cells within the TME undergo senescence and are highly likely to play an important role in tumorigenesis. However, the type of stromal cells that undergo senescence and the role played by cellular senescence in the TME need to be further investigated. Paradoxically, some studies demonstrated that stromal p16 expression inhibits the malignant transformation of preneoplastic lesions [43,47], but others suggested that stromal p16 expression may be involved in tumor invasion [44,45,46,48,49]. Moreover, existing clinical cohort studies have not explored changes in stromal p16 expression during antitumor therapy, so more data are still urgently needed.

## 3. Formation of Senescent Stroma

During the formation of senescent stroma in the TME, three mechanisms are crucial, that is, the influence of external stimuli, the paracrine induction from senescent cells in the TME, and the age-related accumulation of senescent stromal cells. The three mechanisms may coexist to promote the senescence of non-tumor cells in the TME and generate the senescent stroma (Figure 1). However, the transformation is complicated and heterogeneous, so the identification of the initiating step of stromal cell senescence and the detailed molecular mechanisms remain to be elucidated.

Three mechanisms are crucial during the formation of senescent stroma. Firstly, the off-target effect of antitumor treatment is a typical external stimulus which interferes with the physiological function of stromal cells. Other stimuli that cause stromal cell senescence include nutrition dysregulation, microbiota dysbiosis, and so on. Tumor cells in the TME induce cellular senescence of stromal cells mainly through paracrine. In addition, some indirect effects of tumor cells, such as the changes in immune function and hypoxic TME, also contribute to stromal cell senescence. Furthermore, the age-related accumulation of senescent stromal cells is mainly attributed to immunosenescence and increased HLA-E expression that occurs during aging.

### 3.1. The Influence of External Stimuli

The onset of cellular senescence is essentially a cellular defense mechanism [4,9]. Cellular senescence induced by different stimuli differs in the initiation link; for example, DNA damage triggers the ataxia telangiectasia mutated (ATM) or ataxia telangiectasia and Rad3-related protein (AOR) pathways to activate p53 to induce senescence, whereas phosphate and tension homology deleted on chromosome ten (PTEN) loss-induced cellular senescence lacks significant DNA damage and activates p53 mainly through mTOR signaling [2,10,11]. In summary, under the influence of stimuli, non-senescent populations activate the p16/pRB or p53/p21 pathways to transform.

Chemotherapy and radiotherapy are common external stimuli that trigger cellular senescence. This type is termed therapeutic-induced senescence (TIS). Multiple genotoxic agents have been shown to promote cancer cell senescence by causing DNA damage, but few studies have discussed the senescence induction effect of antitumor drugs on stromal cells. Among them, doxorubicin has been studied more extensively. Under the induction of doxorubicin, human umbilical vein endothelial cells and normal epithelial MCF-10A cells underwent senescence and rendered MDA-MB-231 cancer cells more aggressive [52,53]. In two studies, doxorubicin administration induced senescence in mouse embryonic fibroblasts (MEFs) and J774 mouse macrophages via the initiation of the p53/p21 pathway [54,55]. Other antitumor agents, such as telomerase inhibitors (i.e., BIBR15 and GRN163L) and cell cycle inhibitors (i.e., palbociclib and ribociclib) are also known to cause cellular senescence [2]; however, studies on the interaction of these drugs with stromal cells in the TME are still lacking. TIS of stromal cells during and after radiotherapy has also been reported in different malignancies, including skin cancer [56], lung cancer [57,58], and glioblastoma [59,60]. These findings suggest that TIS of stromal cells in antitumor therapy is potentially linked to tumor cell tolerance and also provides insights for the development of novel antitumor therapies. In addition, other factors that cause stromal cell senescence in the TME include nutrition dysregulation [40], age [8,61,62], microbiota dysbiosis with toxin [40,63], and so on. Apparently, cellular senescence induced by abnormal and deleterious stressors is a prevalent defense program of stromal cells.

### 3.2. The Induction from Senescent Cells in TME

The cellular senescence of tumor cells has been widely studied since OIS was first described in 1997 [11]. With the introduction of the concept of the TME, the crosstalk between tumor cells and stromal cells has been extensively researched. As with tumor cells, the senescence of stromal cells is universal, but the temporal sequence of senescence onset of both remains unresolved. 

Tumor cells display an altered secretion profile and result in cellular senescence of stromal cells via the paracrine manner. In contrast to the stimulation of external stressors, the senescence-triggering signal from tumor cells is TME-endogenous. A study confirmed that the expression of the chemokine growth-regulated oncogene 1 (GRO-1, also known as TRIT1) was significantly elevated in ovarian cancer, and the overexpression was shown to be triggered by RAS signaling, resulting in inducing stromal fibroblast senescence, implicating the possibility of tumor-cell-induced stromal cell senescence [64]. Thereafter, studies on mice revealed that senescent tumor cells induce senescence of stromal fibroblasts as well as infiltrating hematopoietic and immune cells [34,36]. The results of the co-culture of stromal cells with tumor cells also facilitate the understanding of cell crosstalk. For instance, one study comparing the effects of three lung cancer histological types on fibroblasts verified that large-cell lung cancer induced normal fibroblast senescence through oxidative stress, whereas adenocarcinoma and squamous cell carcinoma did not [65]. There are multiple types of signaling that may interplay with the activation of p53 and the overexpression of p16. In addition to RAS signaling, TGF-β-directed oxidative stress was identified in a genetically unstable type of oral squamous cell carcinoma (OSCC), a mechanism that enables normal fibroblasts to senesce in co-culture with cancer cells [66]. C-X-C Motif Chemokine Ligand 1 (CXCL1) may also be involved. A study demonstrated that adiponectin, a specific adipocytokine with oncogenic effects, caused tumor cells to secrete CXCL1 to induce stromal cell senescence [67]. The results of this study revealed the interaction of internal and external factors at the TME level to induce stromal cell senescence, reflecting the complexity and systemic nature of cellular senescence-triggering mechanisms. 

Intriguingly, senescent tumor cells and SASP do not appear to be the only way to induce stromal cell senescence, as non-senescent tumor cells have also been demonstrated to possess similar effects. In one study, the conversion of normal fibroblasts to cancer-associated fibroblasts (CAFs) under co-culture conditions with OSCC lines was associated with the regulation of IL6 and CXCL1, with the latter acting in a cancer-specific manner [68]. Intriguingly, CXCL1 was not detected in normal fibroblasts in isolation or OSCC under co-culture conditions, indicating that the induction effect of cancer cells does not necessarily depend on the previously thought functional arm of cellular senescence, SASP, and there may be other mechanisms underlying it. These findings suggest that the existing understanding of the causal relationship between cellular senescence in different cell types in the TME needs to be expanded. Stromal cell senescence may originate from paracrine-induced senescence by the regulation of senescent cancer cells, or it may be a defense reaction by benign non-tumor cells against certain malignant signals in the TME. Confirmation of these conjectures requires additional research to support them.

Moreover, tumor cells also cause senescence of infiltrating immune cells, resulting in the formation of a pro-tumor TME. Most tumors are age-related diseases, and the immunosenescence that occurs during aging plays an integral role in tumor protection [69]. In several studies, it has been verified that in the process of immune senescence, the proportion of partial types of immune cells in TME, such as tumor-associated macrophage and regulatory T cell (Treg), is increased and that these cells induce effector T cell senescence with dysfunction [70,71,72]. In addition, tumor-derived cyclic adenosine monophosphate (cAMP) may act as a molecular messenger to promote T cell senescence in the hypoxic TME [70,71,73]. In a recently proposed hypothesis, immune cells may not only be victims but also accomplices of tumor cells, acting as mediators of tumor cells to trigger stromal cell senescence in TME [27]. This speculation is yet to be proven and is mainly based on findings from recent studies that elucidate the engagement of senescent immune cells in the process of solid tissue senescence [74,75]. In summary, various cell types in the TME exhibit complex interactions with each other, and signals of senescence can be gradually amplified by conveying them to neighboring cells in a paracrine manner.

### 3.3. Age-Associated Accumulation of Senescent Cells

As mentioned earlier, most cancers are age-related diseases. Senescent cells gradually accumulate during aging. Sometimes aging and senescence are confused, but in fact, they are completely different concepts, and most of the hallmarks of the two do not overlap. However, there are commonalities between them, for example, the reprogramming of cell populations in the local microenvironment with aging, especially the increase in the proportion of senescent cells [38,61]. The source of this dynamic change is still unknown, but it is clear that the formation rate of senescent cells is exceeding the elimination rate, leading to senescent cell accumulation [62]. It has been proposed that the accumulation of senescent cells in the TME is attributable to a declining elimination rate, which is implicated in the age-related decline in immune function, i.e., immunosenescence [76]. Consistent with this, a recent study found that senescent skin fibroblasts express the non-classical MHC molecule HLA-E, which interacts with the inhibitory receptor NKG2A, expressed by natural killer and highly differentiated CD8+ T cells, to suppress immune surveillance against senescent cells [77]. Ex vivo comparative experiments also demonstrated that HLA-E expression is increased in senescent cells from skin sections from the elderly compared to the young and in human melanocytic nevi compared to normal skin tissue [77]. A recent study also pointed out that in neuroblastoma, mesenchymal stromal cells in a senescent state can inhibit NK cell activity, thus weakening the immunosurveillance role of NK cells [78]. However, this hypothesis remains controversial, with one study comparing cellular senescence level and contribution in the skin and immune system and arguing that there is no significant association between the two in individuals [79]. Nevertheless, age-associated accumulation of senescent cells most likely results in altered cell populations and extracellular matrix (ECM) in the local microenvironment, ultimately leading to the onset of malignancy [62].

## 4. SASP: A Function Arm of Cellular Senescence and a Complex Network of Signaling Pathways

Senescence is considered a form of cellular “quiescence” due to the “irreversible” stagnation of cell proliferation in senescent cells. However, with breakthroughs in the study of senescent cells, it is becoming clear that senescent cells undergo significant morphological alterations and display an active secretory phenotype, known as SASP, which is the main promoter of the non-cell-autonomous biological effects of senescent stromal cells, locally regulating the TME through the release of various soluble molecules. To clarify the role of senescent stromal cells in the TME, SASP, the downstream functional arm of senescent stromal cells, especially in causing local inflammation, cannot be ignored.

When stromal cells are exposed to various chronic stimuli, the p16/pRB and p53/p21 pathways are activated, triggering a cascade of responses that ultimately induce cellular senescence and various corresponding phenotypes, including SASP. SASP and other phenotypes of senescence (e.g., cell cycle arrest) are decoupled under specific conditions, indicating that SASP production undergoes an independent program, at least in part, of the induction of senescence [80]. The mechanisms that initiate SASP are complex and involve multiple pathways and molecules (Figure 2). Upstream signals triggered by different cellular structure changes at the onset of senescence, such as DNA damage response (DDR), senescence-associated mitochondrial dysfunction, and the degradation of the nuclear lamina protein Lamin B1 [81,82,83,84], induced profound intracellular alterations through distinct cascading reactions, resulting in significant differences in the secretome of the senescent stromal cells [85].

The mechanisms that initiate SASP involve multiple pathways and molecules. Upstream signals mainly include DNA damage response (DDR) and senescence-associated mitochondrial dysfunction. p38MAPK plays an essential role in the induction of SASP. It is activated through pathways initiated by mitochondrial dysfunction and nuclear abnormalities, such as the ATM/TRAF6/TAK1 axis; the kinase is subsequently involved in the PI3K/Akt/mTOR pathway. mTOR is able to activate NF-κB signaling directly or indirectly via IL-1α by regulating its translation. Meanwhile, p38MAPK also controls AUF1 occupancy on SASP mRNAs to maintain their stability. Zscan4 is expressed through the ATM-TRAF6-TAK1 axis during the acute response after DNA damage and translocates to the nucleus. It enables a long-term SASP via NF-κB signaling, forming a positive feedback loop. The cGAS/STING pathway is activated in the presence of cytoplasmic DNA accumulation due to DNase2/TREX1 downregulation. Cytoplasmic chromatin fragments (CCFs) derived from the degraded nuclear lamina protein Lamin B1 are also recognized by cGAS to produce cGAMP and subsequently activate STING to promote SASP expression. GATA-4 links autophagy and DDR to SASP via the expression of IL-1α and TRAF3IP2, an E3 ubiquitin ligase for TRAF6. Most of the pathways above converge to activate the transcription factor NF-κB. Release of the inhibitor IκBα leads to nuclear translocation of NF-κB, resulting in the expression of SASP genes. Activated NF-κB acts on IL1-α, forming a positive feedback loop. The activated transcription factors c/EBPβ increases binding to the promoters of certain SASP genes. Recruitment of transcriptional co-activator BRD4 to super-enhancers adjacent to key SASP genes promotes SASP expression and downstream paracrine signaling. The histone H3-specific demethylase KDM4 remodels chromatin and regulates SASP expression through histone demethylation. (Upstream signals: blue line; NF-κB activation: black line; other genetic and epigenetic regulations: yellow line; transcription and translation: purple line. SASP: senescence-associated secretion phenotype; DDR: DNA damage response; CCFs: cytoplasmic chromatin fragments.)

The three canonical pathways that are currently thought to primarily regulate SASP are p38 mitogen-activated protein kinase (p38 MAPK, also known as MAPK14), CCAAT enhancer-binding protein beta (c/EBPβ), and IL-1α. These pathways induce a variety of typical SASP factors by targeting NF-κb. Several studies have demonstrated that p38 MAPK plays an essential role in the induction of SASP in multiple senescent stromal cells and that it can be activated by pathways initiated by senescence-associated mitochondrial dysfunction and nuclear abnormalities (e.g., ATM/TRAF6/TAK1 axis and ROS) [81,82,86]. This process may involve the engagement of the PI3K/AKT pathway, targeting mTOR in addition to the downstream NF-κb, which is closely associated with chemotherapy-induced SASP factor IL-6 in the context of B-cell lymphoma [87]. In parallel, p38 MAPK was found to remove AUF1 to prevent the latter from binding to the mRNA of SASP factors, thus increasing mRNA stability [88]. c/EBPβ is likewise a proven SASP regulator. During senescence, the full-length activated c/EBPβ isoform LAP2 increases binding to the promoters of certain SASP genes, such as osteopontin (OPN), IL-6, and IL-8 [85]. Another indispensable modulator is IL-1α, whose canonical expression controls a majority of pro-inflammatory SASP factors [80]. IL-1α amplifies the signal by forming a positive feedback loop with NF-κb during the induction of SASP [80]. This process involves mTOR in the translation of IL-1α mRNA, and rapamycin blunts the pro-inflammatory phenotype of senescent cells by interfering with this loop [89].

Yet more findings suggest that the three pathways mentioned above are not the entire spectrum of SASP regulation; they play a crucial role, however, and may only be the tip of the iceberg in a vast network. Unveiling the novel SASP regulators conveys the important message that “one size fits all” in the discussion of cellular senescence and SASP in isolation from induction methods and cell types should be avoided. Several studies demonstrated that in addition to these three pathways, there are other molecules upstream of NF-κb that are activated during senescence [90,91]. The cGAS/STING pathway, an aberrant signaling perceptor of innate immunity, is activated in senescent stromal cells in the presence of cytoplasmic DNA accumulation due to DNase2/TREX1 downregulation and elevates expression of common SASP factors such as IL-1β and IL-6 by promoting ROS production [92]. Another possible cause for triggering the cGAS/STING pathway is the appearance of cytosolic chromatin fragments (CCFs) following the degradation of Lamin B1 [84]. Regardless, GATA-4 links autophagy and DDR to SASP via the expression of IL-1α and TRAF3IP2, an E3 ubiquitin ligase for TRAF6 to regulate NF-κb [90]. NOTCH1 also plays a pivotal role in SASP transcription, and its dynamics mediate changes in SASP factor levels from two distinct secretomes. When NOTCH1 is increased, TGF-β ligands-induced SASP factors are upregulated and partial pro-inflammatory SASP factors are downregulated and vice versa [93]. There are also several other proteins that probably influence SASP factor production at the chromosomal level, e.g., BRD4, KDM4, etc. [94,95]. Unfortunately, whether the altered biological functions and expression levels of these molecules involved are dependent on the stimulus source and tissue type still requires further elucidation compared to canonical pathways in the context of senescence. Whether these pathways are interconnected is also a question that needs to be addressed.

Studies on these molecules and pathways are mainly based on the classical soluble SASP factors. Strikingly, recent investigations showed that small extracellular vesicles (sEVs) are also biological effect mediators of senescent stromal cells, also known as evSASP. sEVs released from senescent stromal cells display distinct size distribution [96]. Few clues indicate that interferon-induced transmembrane protein 3 (IFITM3) and NF-κb participate in the modulation of sEVs [97,98]. At this stage, however, little is known about the concrete mechanisms by which these sEVs are produced. The only certainty is that evSASP plays an active role in the regulation of the TME, as do the soluble SASP factors [96,97,98]. A more in-depth exploration of the mechanisms of production of various SASP factors would help to map a more systematic and complete upstream network of SASP regulation to better understand the role played by senescent stromal cells. 

## 5. How Senescent Stroma Promotes Tumor Progression

Decades ago, Krtolica et al. demonstrated the capacity of senescent fibroblasts to promote tumorigenesis, indicating cellular senescence as a case of evolutionary antagonistic pleiotropy that, despite restraining tumor growth in the early stages, may nevertheless exhibit pro-tumorigenic effects when senescence occurs in benign stromal cells [26]. Emerging evidence supports the role of senescent stromal cells as an accomplice in the growth of a variety of epithelial-derived solid tumors [52,99,100]. Similarly, recent studies have shown that senescent mesenchymal stromal cells contribute to the development of myeloid tumors [101,102,103,104]. The positive effect of the senescent stroma on tumor progression has been well established, but behind the scenes are the molecular mechanisms that link senescent stroma, cancer cells, and the TME.

As we described earlier, senescent stromal cells release SASP factors into the TME, thereby affecting malignant cells and ECM. SASP is the linchpin in understanding the tumor promotion by senescent stromal cells. SASP factors induce and enhance tumor cell senescence in both paracrine and autocrine ways to arrest tumor cell proliferation [21,36]. Ironically, numerous SASP factors promote tumor growth and migration in very different contexts [52,105,106,107], ultimately paving the road to hell. Here, we will discuss the main mechanisms by which SASP promotes tumor progression.

### 5.1. Epithelial–Mesenchymal Transition (EMT)

EMT refers to the cellular process of reversible transformation of epithelial cells into mesothelial cells and plays a momentous role in oncogenesis [108]. When epithelial-derived malignant cells receive signals from the TME, the expression of many genes is activated or inhibited by epigenetic modifications, leading to a phenotypic conversion [109]. During EMT, the expression of epithelial cadherin (E-cadherin), a component responsible for adhesion junctions in epithelial tissues arranged regularly with originally apical-basal polarity, is suppressed, and the typical polygonal, cobblestone morphology of epithelial cells is gradually lost, progressively changing to a spindle-shaped mesenchymal morphology [109,110]. Ultimately, cancer cells that undergo EMT acquire a potentially more malignant phenotype and exhibit an enhanced ability in proliferation, invasiveness, and resistance.

EMT is a non-cell autonomous process, meaning that the initiation of the EMT program requires modulation from external contributors. Hypoxia and external agents in the TME are critical signals, while molecules secreted by various active stromal cells, including senescent stromal cells, are also known sources that elicit EMT [110]. Senescent stromal cells release SASP factors to induce EMT of cancer cells via paracrine secretion. The signaling pathways that have been shown to robustly initiate the EMT program are TGF-β, WNTs, NOTCH, and mitogenic growth factors [109], and multiple studies support the involvement of SASP factors produced by senescent stromal cells.

A study evaluating the role of senescent peritoneal mesothelial cells (HPMCs) in colorectal cancers found that conditioned medium (CM) of senescent HPMCs promoted the colorectal cancer cell line SW480 to undergo EMT, closely associated with stromal-derived TGF-β1 [111]. In the context of hepatocellular carcinoma, upregulation of TGF-β and EMT promotion in cancer cells were also observed following hepatic stellate cell senescence [112]. Other SASP factors that facilitate EMT in cancer cells include the pro-inflammatory factors IL-6 and IL-8 [112,113], Serine protease inhibitor Kazal-type 1 (SPINK1) [106], amphiregulin (AREG) [107,114], epiregulin (ERPG) [115], and WNT16B [114]. These SASP factors are secreted in the context of DNA damage and are directly or indirectly implicated in the regulation of the cancer cell EMT program.

Following the interaction of SASP factors with cell surface receptors, specific intracellular molecules are activated to initiate the EMT procedure. The SMAD signaling pathway plays a vital role in EMT development. TGF-β binds to a complex of TGF-β receptor type 1 (TGFβR1) and TGFβR2 on the cell surface, which in turn activates SMAD2 and SMAD3, the latter two forming a trimer with SMAD4, which enters the nucleus as a transcription factor to regulate the expression of EMT-transcription factors [109,116]. In addition, several studies have shown that SASP factors activate non-SMAD pathways to promote EMT; for example, IL-8 initiates the JAK2-STAT3-SNAIL pathway in the context of hepatocellular carcinoma, and AREG activates EGFR, which in turn initiates the PI3K/Akt/mTOR and MAPK pathways to induce EMT in the context of prostate cancer [107,112]. Yet more studies have only confirmed the induction of EMT by SASP factors, and the detailed mechanisms have not been explored in depth.

The expression of mesenchymal cell biomarkers is progressively increased following EMT induction. The altered cell phenotype is accompanied by reduced E-cadherin expression and disruption of cell junctions, facilitating cancer cell migration [109]. Intriguingly, EMT is also frequently observed to increase cancer cell stemness, promote angiogenesis, and remodel ECM, protumor effects that are induced by senescent stromal cells and will be discussed subsequently. This demonstrates the complexity and magnitude of the function of stroma-derived SASP factors in neoplasia regulation.

### 5.2. Cancer Stem Cells (CSCs) and Cancer Stemness

Cancer cell populations are heterogeneous, with different phenotypes of cancer cells differing significantly in terms of proliferation, invasiveness, and tolerance. In this respect, CSCs refer to a subset of the cancer cell population with major properties including self-renewal, clonal tumor initiation capacity, and clonal long-term reproductive potential [117]. Two main viewpoints existed in the past regarding the provenance of CSCs. The hierarchical model suggests that CSCs are derived from stem cells that evade surveillance and undergo a malignant transformation, and that this particular population generates short-lived offspring through continuous self-renewal, similar to the biological behavior of stem cells [118,119]. The other model, the stochastic model, suggests that every cancer cell shares the opportunity equally to convert into CSCs and participate in promoting tumorigenesis [117,120]. Whereas the proposal of CSC plasticity reconciles the two theories, differentiated cancer cells receive specific signals from the adjacent microenvironment and experience dedifferentiation back into the CSC pool, which may be driven both by an innate genetic profile (the hierarchical theory) and by stem cell-like permissive epigenetic modifications (the stochastic theory) [117,121].

Having understood the generation of CSCs, another question is the driving force behind the incremental increase of CSCs in the cancer cell population, or what factors lead to the conversion of non-CSCs to CSCs. The TME, as the extrinsic environment of cancer cells, is an asset in the modulation of cancer cell plasticity [121,122]. Colorectal cancer models underpin research to understand the mechanisms. The process of colorectal cancer establishment is complemented by the activation of NF-κb and the constant stimulation of inflammation, which is reminiscent of biological features of senescent stromal cells and SASP factors [121]. Indeed, there is growing evidence that senescent stromal cells release SASP factors, which cause non-CSCs to dedifferentiate and transmute into more malignant CSCs.

As staple members of the SASP factors, the proinflammatory IL-6 and IL-8 are recognized for their role in the generation of CSCs. Breast cancer models provide support for understanding the interaction between senescence-related inflammation and cancer stemness. Kim et al., in an elegant experiment, furnished direct testimony that IL-6 regulates stemness-associated gene OCT-4 activity in differentiated cancer cells via the JAK1/STAT3 pathway [123]. A study confirmed the vital impact of environmental selection represented by the TME on cancer stemness promotion. A trastuzumab-tolerant breast cancer model was constructed by knocking out the PTEN gene in HER2 overexpressing breast cancer cell lines to simulate the environment of long-term trastuzumab administration, demonstrating upregulation of proinflammatory factor expression in the context of PTEN deletion and eventual expansion of the CSCs population through the IL-6 inflammatory loop [124]. Further studies have shown that treatment with IL-6- or IL-8-enriched senescence CM induces a self- and cross-reinforced senescence/inflammatory milieu, rendering the otherwise less aggressive MCF-7 breast cancer cells stemness-enhanced [113], which was consistent with another study concerning the effect of IL-8 on MCF-7 breast cancer cells [125]. In lieu of breast cancer, one study, in the context of colon cancer, demonstrated that IL-8 targeting of CXCR2 facilitated the orientation of human-bone-marrow-derived mesenchymal stem cells towards the CSC population, thereby fostering the creation of a niche in favor of CSCs [126]. Unfortunately, the above reports focus on the effects of IL-6 and IL-8 on cancer cells in terms of stemness promotion and malignant phenotypic alterations, while it remains unproven whether IL-6 and IL-8 production is associated with senescent stromal cells. A report revealed that myofibroblast-derived IL-6 and IL-8 activate the NOTCH/HES1 and STAT3 pathways to enhance cancer stemness in colon cancer [127]. Considering the nature of the overlap between CAFs, senescent fibroblasts, and myofibroblasts (see BOX1), this could be perceived as a compelling argument for the expansion of the CSCs population by senescent stromal cells via proinflammatory SASP factors. The impact of miscellaneous SASP factors on the improvement of stemness remains to be investigated. A study illustrated that SPINK1 reprograms cancer cell transcriptome-wide expression to promote EMT and CSC growth [106]. However, whether senescent stromal cells enhance cancer stemness by producing pro-inflammatory SASP remains controversial, and this void requires support from additional experimental evidence.

Another reason to propose that SASP promotes stemness is that blocking NF-κb, the major SASP regulator, remarkably reduces stemness [128,129]. This raised new thinking about whether blocking NF-κb to inhibit the expansion of CSCs is relevant to preventing senescence amplification mediated by paracrine signals in the TME [129]. As previously discussed, cancer cells are also susceptible to senescence, and cellular senescence is an integral cellular program that limits tumor progression. Moreover, numerous pieces of evidence give the stereotype that the tumor-promoting effects of cellular senescence are associated with bystander effects, which interfere with the physiological function of stromal cells, whereas the senescence of tumor cells themselves is anti-neoplastic. Intuitively, the features of senescent cells and stem cells are not compatible. However, the discovery by Milanovic et al. provides an insight into the fact that senescent cancer cells are still allowed to return to the cell cycle and dedifferentiate into CSCs to devastate more severely [7]. A possible explanation is that under selective pressure, cancer cells undergo senescence and selectively fit clones, i.e., cancer stem cell populations grow and spread, acquiring more aggressive tumorigenicity and metastasis. Considering the contribution of exogenous drugs and toxicants to both bystander cell senescence and tumor stemness elevation [128,130], it is tempting to rethink the link between senescence and stemness and the role that SASP-mediated paracrine senescence plays in both. It is currently recognized that the presence of senescence-associated stemness is a mechanism inherent in the evolutionary process to cope with stressful damage and ironically confers on tumor cells superior survivability in hostile conditions [7]. Yet these results were accomplished with artificial intervention; to better validate the experimental results, spontaneous models are required to further gauge the effects of spatiotemporal factors and guide a better understanding of the relationship between cellular senescence and cancer cell stemness.

### 5.3. Angiogenesis

Angiogenesis refers to the establishment of new blood vessels from pre-existing vessels [131]. Angiogenesis is vital to tumor growth, with new capillaries improving the degree of hypoxia and transporting nutrients and metabolites for the tumor. A variety of pro- and anti-angiogenic forces are present in the TME; they interact to determine angiogenesis activity within the tumor [132]. Tumor vascularization is initiated when pro-angiogenic forces are predominant, a process called “the angiogenic switch” [133]. Of all the pro-angiogenic forces, without a doubt, angiogenic factors are predominant. The three most recognized angiogenic factors are the vascular endothelial growth factor (VEGF), the fibroblast growth factor (FGF), and the platelet-derived growth factor (PDGF) [131]. A multitude of cytokines secreted by senescent stromal cells, meanwhile, crosstalk with angiogenic factors and thus impinge on the angiogenic switch.

Cultivating three different types of breast cancer cell lines by CM derived from young and senescent HPMCs, it was observed that the secretory levels of pro-angiogenic agents, including CXCL1, CXCL8, the hepatocyte growth factor (HGF), and the VEGF, were significantly increased in cancer cells [134]. Senescent HPMCs were proven to modulate cancer cells by secreting IL-6 and TGF-β1, promoting tumor vascularization through HIF-1α, NF-κb/p50, and AP-1/c-Jun pathways [134]. A subsequent article demonstrated that senescent populations in ovarian cancer cells induce normal HPMC senescence via paracrine secretion to cause stromal cell secretome reprogramming, ultimately resulting in a vicious cycle, offering an understanding of the spontaneous process of de novo tumor development [135]. These senescent immune cells in the TME not only upregulate the secretion of the pro-angiogenic agents matrix metalloproteinase (MMP)9, VEGF-A, and IL-8 but also reduce the synthesis of IP-10 (also known as CXCL10), an important angiogenesis inhibitor, thereby triggering the “angiogenic switch” [136]. Altogether, these studies gave support to the role of senescent stromal cells in the facilitation of angiogenesis.

### 5.4. ECM Remodeling and MMPs

As a non-cellular component of the TME, the ECM deposited by fibroblasts, is critical to the maintenance of tissue integrity, while ECM remodeling, which involves basement membrane disintegration, is universal in the maintenance of physiological homeostasis and abnormal pathological alterations [62]. In the tumor context, ECM remodeling is tightly linked to tumor proliferation, angiogenesis, and distant metastasis.

Stromal-derived SASP factors are involved in ECM remodeling. Among them, the most essential group is MMPs. The upregulation of MMPs in senescent fibroblasts was extensively reported. One of the main models applied to study and understand age-related chronic inflammation is the aged-skin model [137]. In aged skin, MMPs cause disruption of tissue homeostasis by degrading the ECM. It has previously been shown that senescent fibroblasts promote early tumor growth by secreting MMP1 (interstitial collagenase) and MMP2 (72kDa type IV collagenase), which modulate PAR1 in malignant cells via a paracrine manner [138]. Further investigations proved that MMPs and PAR1 are upregulated in aged skin compared to young, healthy samples [138]. More studies revealed that tumor invasion facilitated by MMPs is not limited to skin cancer. Bleomycin-induced senescent fibroblasts stimulate early growth of MDA-MB-231 cells (breast cancer model), and administration of the MMP inhibitor GM6001 reversed this effect [139], which was consistent with ionizing-radiation-induced senescent human lung fibroblasts in the context of lung cancer [140]. In addition, MMPs also affect cancer prognosis. In contrast to the better prognosis subtype, the genetically stable OSCC, CAFs derived from genetically unstable OSCC exhibited upregulation of MMP2 and a corresponding greater contribution for ECM destruction and keratinocyte discohesion [141]. Senescent HPMCs promote peritoneal metastasis in colorectal cancer, and the enhancement of tumor aggressiveness is due to increased expression levels of several SASP factors, including MMP-3 [111]. Altogether, SASP factors, especially MMPs contribute to tumorigenesis and poor prognosis via ECM remodeling, which is another convincing piece of evidence for the detrimental effects of senescent stromal cells.

## 6. Distant Metastasis: Distinct Niches Shaped by Senescent Stroma

Following early growth, cancer cells break away from the primary lesion and metastasize to distant sites through blood vessels, lymphatic vessels, and coeloms. The TMEs of primary and metastatic lesions are distinct, and therefore, the conventional concept of the TME does not describe the micro-ecology of the lesion completely and accurately [25]. Laplane et al. proposed a new concept, the tumor organismal environment, to distinguish microenvironments at different distances from the primary lesion, which was used to compensate for the shortcomings of the original TME theory [142,143]. To be sure, there is variation in niches of the TME, which is an essential factor contributing to tumor microscale heterogeneity.

For certain cancers, there is a preference for distant metastatic sites. For example, the peritoneum is a favored site of metastasis for numerous solid malignancies. In addition to the close association between peritoneal implantation and natural anatomic structures, senescence-related stroma changes in the peritoneal microenvironment are also influential. A study showed that malignant ascites upregulated HGF and GRO-1 to induce senescence of normal HPMCs, the latter by releasing multiple SASP factors to stimulate metastatic cancer cell adhesion, proliferation, and migration [105]. Further investigations extracted surgical specimens from ovarian cancer patients with peritoneal metastases and discovered that senescent HPMCs were distributed around the cancer cells, indicating that the senescence induction of HPMCs by cancer cells is not only through soluble factors but also partially involves cell–cell contact [144]. Similar observations were also detected in the context of colorectal cancer [111]. To some extent, these indications answer the contribution of the senescent stroma to the directional metastasis of cancer cells, although the chronology and causality remain controversial. The effect of senescent stroma on distant metastases is, of course, not confined to the peritoneal cavity. Senescent osteoblasts increase the activity of osteoclasts and create fertile seeding areas in bone for breast cancer cells by causing matrix-associated alterations [145]. This theory is also supported by a recent study in which Fane et al. showed through sophisticated experiments that in aged animals, the aged lung stroma is more conducive to malignant melanoma growth than the skin, thus explaining to some extent the propensity of malignant melanoma to metastasize to the lung [146]. Altered secretion of soluble substances by aged lung fibroblasts provides a suitable environment for malignant melanoma cell growth through the WNT pathway [146,147]. Considering that the accumulation of senescent fibroblasts is one of the aging changes, whether cellular senescence is involved is a topic of concern.

In summary, the current evidence suggests that the senescent stroma shapes the unique microenvironment and niches in distant tissues in a way that is distinct from the physiological condition, providing a breeding ground for directional metastasis and seeding for cancer cells. Nonetheless, much work remains to be completed to improve the understanding of the relationship between senescent stroma and distant metastasis. For instance, whether the senescent stromal-cell-derived SASP factors induce cancer cells for targeted migration and seeding to favored distant sites is still unclear.

## 7. Senescent Stroma and Immune Infiltration in the TME

The TME exhibits two hallmarks: hypoxia and immune infiltration [25]. Infiltrated immune cells are an integral part of the TME, and these populations are closely linked to tumorigenesis, distant metastasis, and prognosis [25,148]. It was previously believed that immune cells infiltrating the TME interact with cancer cells to limit tumor growth, but accumulating shreds of evidence challenge this view [149]. Several studies have shown a complex and intimate association between the pro-inflammatory effects exerted by several SASP factors and immune cell infiltration. Herein, how senescent stroma potentially affects immune infiltration will be discussed.

### 7.1. Inflammation Induced by SASP

One of the hallmarks of aging is an increase in systemic low-grade chronic inflammation, a process known as “inflammaging” [62]. Age-related changes of many cytokines are thought to be associated with chronic inflammation, including IL-1, IL-6, IL-lα, IL-1β, IFNγ, etc., which are recognized as typical SASP factors [150]. It is worth noting that although age and senescence are not the same concept, aging stroma does accompany the onset of cellular senescence and SASP factor release, thereby making cellular senescence a bridge between aging, chronic inflammation, and tumorigenesis [123,151].

Inflammasome, a multiprotein complex comprising caspase 1 and multiple adapter molecules, is tightly linked to the pathogenesis of inflammation. It is a platform for the processing of several interleukins, with specific subsets of the latter being key SASP regulators [36,152]. Through elegant experiments, Acosta et al. showed that SASP is regulated by a sophisticated program governed by inflammasomes, suggesting a causal relationship and the role of SASP in the onset of inflammation [36]. This finding was further confirmed by the observation that in pancreatic cancer, blocking the IL-1 pathway decouples SASP from other senescence-related phenotypes, thereby inhibiting macrophage recruitment in the TME and desmoplastic tissue generation, a hallmark of pancreatic cancer [80].

Multiple inflammatory cells are the bridge between stroma-derived SASP factors and numerous types of cancer. Neutrophils are recruited to the TME in the presence of high endothelial expression of N1ICD, the intracellular domain of NOTCH, and thus participate in the inflammatory response [153]. Meanwhile, robust expression of N1ICD has been demonstrated in a variety of solid tumors, notably in melanoma and metastatic samples, and is involved in the induction of endothelial cell senescence [153]. Another study identified that ROS derived from senescent fibroblasts, a non-protein molecule in SASP factors, mediated the neutrophil recruitment in the context of acute liver injury and that neutrophils induced senescence in a paracrine manner through telomere dysfunction, implying that the crosstalk of senescent cells and infiltrated neutrophils build a positive loop [154]. In addition, macrophages, a component of innate immunity, are similarly recruited by SASP factors released from the senescent stroma, triggering an inflammatory response in pancreatic cancer and adipose tissue [80,155]. Furthermore, senescent T cells, characterized as CD8^+^CD27^−^CD45RA^+^, have also been shown to participate in age-associated inflammation [86,136].

Numerous models with a tumor context demonstrate a direct link between tumor development and pro-inflammatory SASP-factor-related signaling. Unfortunately, few of them are available to confirm a causal relationship between many of these components and age-related cancer progression. Supplementary research is necessary to clarify the intricate mechanisms by which senescence regulates the transition from inflammation to tumor initiation.

### 7.2. Immunosuppressive Cells: Assistant to the Progression of Malignancy

Immunosuppressive cells are regarded as a pivotal promoter of tumor progression. The infiltration of these populations in the TME is involved in the cancer cell immune evasion via manifold pathways and may answer, to some extent, the specific mechanisms underlying the switch from chronic inflammation to cancer [156]. In recent years, emerging evidence has demonstrated that senescent stromal cells create an immunosuppressive TME by inducing immunosuppressive cell infiltration and modulating immune cell function, rendering the TME a sanctuary for the devil.

Myeloid-derived suppressor cells (MDSCs) are one of the immunosuppressive populations recruited by senescent stroma. A previous publication showed that CDK4/6 inhibitor-induced senescence in fibroblasts promoted tumor growth after co-injection with different melanomas cell lines, with a significant increase in MDSCs infiltration, which provides an additional growth advantage to the tumors [99]. However, the concrete mechanism of chemotaxis enhancement of MDSCs is not well understood and may not be relevant to SASP. A study revealed that in monocytic MDSCs, upregulation of senescence regulatory molecules p16 and p21 inhibits CDKs-mediated phosphorylation and inactivation of SMAD3, resulting in high expression of the chemokine receptor CX3CR1, culminating in enhanced tumor growth via the CX3CL1/CX3CR1 axis [35]. Intriguingly, the upregulation of p16 and p21 expression in these MDSCs was not accompanied by other senescence-related signatures, such as DNA damage signs, decreased Lamin B1 expression, and IL-6 induction, indicating that these cells may not be in the state of cellular senescence. However, senescent fibroblasts and premalignant hepatocytes recruit MDSCs through IL-6 and CCL2-mediated pathways to produce immunosuppressive and tumor-permissive TMEs, respectively, which reveals a synergistic effect between the establishment of the senescent stroma in the TME and the chemotactic movement of MDSCs [157,158].

An additional population that potentially participates in immunosuppressive TME establishment is monocytes and macrophages (noted as Mo/Ma). Through Tim-3/galectin-9 (Gal-9) and CD40L/CD40 axes, tumor-induced senescent T cells interact with Mo/Ma, participating in the latter’s canonical activation and thus indirectly facilitating angiogenesis [136]. In another study, H-Ras-treated aged skin was accompanied by strong activation of Toll-like receptor (TLR) and NF-κb signaling and significant enrichment of immune cells [38]. Similar findings were obtained in the context of liver cancer [91]. Further observations found intense activation of IL-4 and IL-10, triggering downstream JAK/STAT axis to activate T helper 2 (Th2) cells and thus repress inflammation while recruiting ECM regulatory cells such as macrophages, shaping a tumor-favored TME [38]. Intriguingly, Lujambio et al. identified that p53-expressing senescent liver stellate cells released cytokines that induced macrophages to enter the tumor-suppressive M1 state and inhibited macrophages from accessing the pro-tumor M2 state [159], which was contrary to the findings of some other publications [40,91]. This may be due to differences in the onset of fibrosis in liver tissue under different administrations, leading to alterations in the biological role of senescent hepatic stellate cells.

Senescent stromal cells are involved in shaping the immunosuppressive TME, although in partial circumstances, they also exhibit antitumor effects. A question worth exploring but remains unresolved is that there are both pros and cons to senescent stromal cells, and it is interesting to consider which populations need to be removed.

### 7.3. Senescent T Cells

T cells are the main effector population in which the immune system functions. During aging and immunosenescence, T cells are significantly altered in quantity and quality under the influence of age-associated inflammation, SASP, and thymic degeneration [69]. T cell senescence is a major hallmark of immunosenescence and one of the main forms of T cell dysfunction in cancer [69]. Not all T cells undergo senescence, and those that undergo relevant phenotypic changes are mainly terminally differentiated and EMRA cells, effector memory T cells that reacquired naive-like state [69,70]. Senescent T cells display a variety of characteristic changes, including the loss of CD27 and CD28, the alterations in the immune checkpoint-related molecule expression (e.g., Tim-3, PD-1, TIGIT), and the elevated expression of senescence markers such as p16, p21, and p53 [37,136,160,161,162]. Moreover, following senescence, T cells manifest a unique pro-inflammatory secretome [86,161] and simultaneously adjust in the metabolic profile, possessing activated glucose metabolism and imbalanced lipid accumulation [163]. 

The induction of T cell senescence is a biological process with intricate mechanisms. Several earlier studies illustrated that normal T cells obtained a senescence phenotype under co-incubation conditions with tumor cells [37,136]. Senescence of infiltrating T cells is triggered similarly to other stromal cells, but other signals are probably engaged. Immunoglobulin-like transcript 4 (ILT4), an immunosuppressant expressed by malignant cells, was demonstrated to be implicated in T cell senescence induction in the TME [164]. Treg is also a culprit. Treg, in crosstalk with responder T cells, generates metabolic competition through the AMPK-associated glucose utilization pathway, thereby invoking DNA damage of responder T cells in TME and initiating senescence-related signals [71].

Following the acquisition of senescence phenotypes, senescent T cells undergo function adjustments. A study identified the most characteristic and heterogeneous subset of senescent CD8^+^ T cells, characterized as CD8^+^ CD45RA^+^ CD27^−^EMRA [86]. This population of T cells secretes SASP factors under the control of p38 MAPK and is thus culpable in age-related inflammation. Moreover, enhanced T cell ROS and NO released and altered the level of immune checkpoint-related molecules after senescence mediates NF-κb-dependent Mo/Ma canonical activation and angiogenesis promotion, thereby reshaping the TME [136]. A survey revealed a general upregulation of TIGIT level during CD8^+^ T cell senescence [161]. Further exploration uncovered that T cell inhibitory receptors including CD160, 2B4, and PD-1 expression were elevated, unveiling that the essence of CD8^+^ T cell senescence is over-activation leading to depletion [161]. Intriguingly, senescent T cells gradually lose T cell antigen receptor signaling activity but retain innate-like killing activity [165]. Supplementary studies demonstrated potential mechanisms by which sestrins reprogram the expression profile of non-proliferating T cells during senescence and promote the synthesis of the NKG2D-DAP12 complex, which binds to NKG2D ligand-positive cells to exert cytotoxic effects [165]. Overall, these works jointly describe the cellular changes and function adaptations of T cells undergoing senescence and provide a tremendous contribution to the recognition of the role played by senescent T cells in tumor immunity.

## 8. Tumor Therapy Resistance Promoted by Senescent Stroma

One of the main issues in cancer treatment at the present stage is the tolerance of neoplastic tissue to radiotherapy and chemotherapy. Several therapy strategies are relatively effective in the very early stages of application, but resistance develops over time, leading to eventual failure. An obvious explanation is that, in contrast to small neoplastic tissues cultured in vitro and in animal models, clinically encountered neoplasms harbor a complex TME accompanied by diverse niches [25]. In the course of radio-chemotherapy, various components of the TME may sustain damage and develop phenotypic and functional modifications, ultimately evolving as a shelter for cancer cells [148].

As previously outlined, cellular senescence could be triggered by various cellular-level injuries due to radio-chemotherapy. Inevitably, when irradiation or chemotherapeutic agents are delivered to localized lesions, bystander cells are damaged and TIS is induced, presenting a so-called bystander effect or off-target effect [166,167]. Eventually, the phenomenon of cellular senescence spreads throughout each compartment of the TME, forming a senescent stroma. The synergistic effects of senescent stroma on neoplasm growth are convoluted, but in general, there are two aspects, direct tumor progression promotion (EMT, CSC generation, angiogenesis, ECM remodeling, distant metastasis, etc.) and indirect influence on tumor immunity (inflammation promotion, immunosuppressive cell recruitment, T cell senescence induction, etc.), the detailed evidence of which has been initially discussed above.

The observation that stromal cell TIS elicits therapy failure is reasonably definitive. Palbociclib is compelling evidence. As an FDA-approved agent for the management of progressive breast cancer, palbociclib functions as a CDK4/6 inhibitor to govern the cell cycle [9]. Though the drug was not originally invented as a senescence-inducing agent, it does display senescence-triggering effects. Palbociclib-exposed stromal cells confer immunosuppression features by the TME for chemotaxis of MDSCs [99]. In addition, several cytotoxic drugs commonly utilized in oncotherapy such as doxorubicin, mitomycin, and bleomycin are also tools to recognize how the senescent stroma builds therapeutic resistance [87,115]. As the area continues to progress, the emphasis on senescent stroma has surfaced. Stromal cell damage and senescence, if properly tackled, could be the Achilles’ heel of neoplasms [168,169]. A novel therapy strategy, “one-two punch” therapy (see BOX2), is likely to be the leading research direction of the next phase.

## 9. Prospects

The existing studies and evidence summarized clearly show that benign stromal cells in the TME are not only victims of external damage but also emerge as focal contributors to tumor progression, modulation of immune function, and tolerance development after stress-induced cellular senescence and are underlying accomplices in tumor establishment and advancement. The accumulation of senescent cells in the TME creates a breeding ground for tumor preferences in the specific context of organismal conditions and pathogenesis. Of particular alarm, factors such as age-related changes and antitumor therapy are major sources of cellular senescence induction, implying that senescent stroma formation is quite common in patients suffering from cancer. If the senescent cells are eliminated by orderly medication, there might be hope for successful treatment. In this respect, senolytic drugs appear to be a promising Trojan horse to break through the city of Troy; however, the danger of these drugs themselves to the intact organism is still unknown and these difficulties have not yet been encountered in cell and animal models at current stages. There is still a long way to go before the clinical application of research findings in the area of cellular senescence can be incorporated into oncotherapy.

## 10. Conclusions

To our best knowledge, few prior articles reviewed the effects and detailed mechanisms of senescent stroma on the TME. This article describes the discovery and formation of senescent stroma and SASP, focusing on the role of senescent stroma in the TME, including promoting tumor progression and distal metastasis, modulating immune infiltration, and establishing tumor therapy resistance. Finally, it concludes with an outlook on future prospects. However, there are considerable issues that remain to be tackled, including the altered function of senescent stromal cells, the specific signaling network of tumor-cell-induced stromal cell senescence, and the effects of cellular senescence in spontaneous cancer models. Collectively, this article reviews the role and molecular mechanisms of cellular senescence in the TME, discusses how cellular senescence promulgates tumors, and hopefully provides inspiration for the study of cellular senescence and cancer.

### 10.1. BOX1 CAFs, Myofibroblasts, and Senescent Fibroblasts: Unresolved Dispute

CAFs are a subset of fibroblasts that possess the ability to promote malignant tumor progression [170]. The definition of CAFs is still controversial and the complex molecular characteristics and corresponding heterogeneity of CAFs remain a major obstacle to understanding the role of CAFs in cancer. At the current stage, it is generally accepted that the activation of CAFs upregulates α smooth muscle actin (αSMA) as a marker to assess the quantity and status of CAFs, which is also one of the biomarkers of myofibroblasts [171]. Myofibroblasts have been shown to foster tumor metastasis and impact prognosis [172,173,174,175]. However, the two cell populations, myofibroblasts and CAFs, are not exactly equivalent. CAFs appear to have a more robust role in facilitating tumor aggressiveness, and there are discrepancies in the capacity to deposit ECM [176]. Likewise, CAFs and senescent fibroblasts share multiple characteristics, with a high degree of overlap between the two in terms of the secretome [176,177]. Simultaneously, a study also showed that the tumor-promoting effect of myofibroblasts was associated with IL-6 and IL-8, the two major SASP factors [127]. It is obvious that the three share a number of common features and that there may be confusion in some of the literature, which requires more research work to address.

### 10.2. BOX2 “One-Two Punch” Therapy

Considering that the development of drug resistance may result in therapy failure, strategies for drug combination therapy have been developed clinically. If the right type of drug is selected, then combination therapy targeting specific weaknesses in the lesion will significantly improve outcomes [178]. A large factor restricting combination therapy is the underlying toxicity, which may contribute to poor prognosis [179]. Thus, the theory of “one-two punch” therapy was introduced, whereby the first drug induces senescence of cancer cells to generate a vital defect, and then a second drug is used to target this artificial Achilles’ heel. To ensure that the “one-two punch” therapy is sufficiently effective, the first drug-induced acquired frailty should be stable [9]. Cellular senescence fits into this theory because it possesses adequate stability as a cellular state activated by external stress [180]. Furthermore, the significant alterations in the metabolic profile, secretome, and gene expression of senescent cells represent a promising target for antitumor agents. Overall, the “one-two punch” is essentially a clinical strategy in which senescence-inducing drugs trigger cellular senescence, and then senolytics are used to selectively eliminate senescent cells [16].

## Figures and Tables

**Figure 1 cancers-15-01927-f001:**
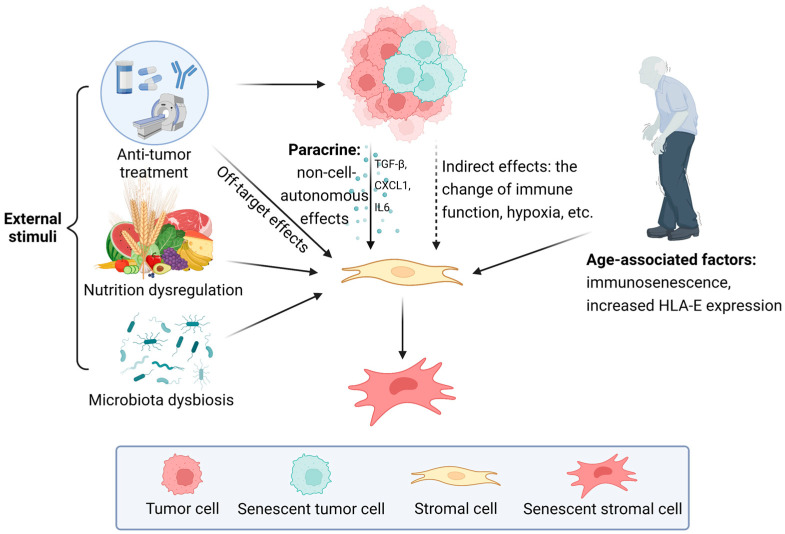
Mechanisms involved in the formation of senescent stroma. (Created with BioRender.com).

**Figure 2 cancers-15-01927-f002:**
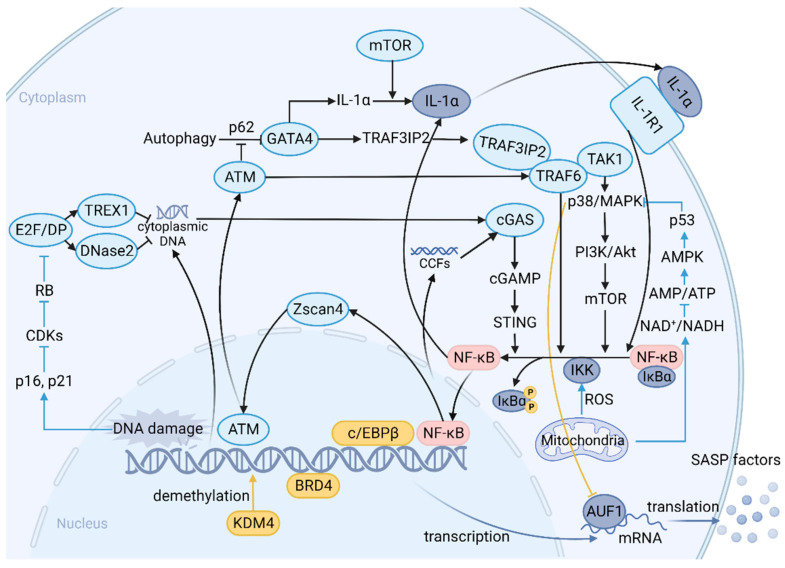
Mechanisms involved in senescence-associated secretion phenotype (SASP) regulation. (Created with BioRender.com).

## Data Availability

The data can be shared up on request.

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
