# Peer review of "Senescent Stromal Cells in the Tumor Microenvironment: Victims or Accomplices?"

_cancers, 2023, doi:10.3390/cancers15071927_

Round 1
Reviewer 1 Report
Manuscript entitled ‘Senescent stromal cells in the tumor microenvironment: victims or accomplices’ is a well written manuscript, useful and extremely inspiring. Moreover, this manuscript certainly contributes to the deepening the understanding of the impact that cellular senescence has on tumor microenvironment.
The minor comments:
· Affiliations should be indicated in accordance with the marking next to the author's name
· Check the spelling on figure 1
· Section 7, at the end of the first sentence a word is missing (I assume ‘infiltration’)
Author Response
Reviewer 1
The minor comments:
- Affiliations should be indicated in accordance with the marking next to the author's name
Re: Thank you for the reminder. We have amended the error in the marking of affiliations.
- Check the spelling on figure 1
Re: The error has been amended as requested.
- Section 7, at the end of the first sentence a word is missing (I assume ‘infiltration’)
Re: We apologize for the omission in the writing process, and have now filled in the missing word.
Reviewer 2 Report
Here authors review the role of stromal cell senescence and their SASP on overall tumor growth, invasiveness, metastasis, immune escape, and treatment-resistant mechanisms. Overall this is a really good discussion on the potentially important aspects of stromal senescence and the authors did a good job of systematically discussing the factors and signaling pathways that drive senescence in the tumor stroma and how it affects various mechanisms in TME leading to the outcome in favor of tumor growth and metastasis.
Below are some points that need attention in order to improve the overall quality and readability of this review.
Major comments.
1. Figure 2 needs to modify in a way to simplify the interactions among different pathways that trigger SASP production. There are many interactions described in the legend/ text that do not reflect in the figure, for example, mTOR's ability to activate NFkB directly or in association with mTOR/IL-1R1 signaling did not effectively communicate in the figure. To clearly denote the path of key interacting axes that trigger the final event (NFkB activation, cEBPb binding, or other genetic and epigenetic change), the author can make use of different colors or numbers, so that readers can easily comprehend.
2. What advantage does cancer cell-intrinsic SASP and stromal-derived SASP has on overall tumor progression? Does the senescence of cancer cells result from selection pressure of immunity, therapy, nutritional resources, etc., resulting in the growth and proliferation of selectively fit clones that gain more aggressive tumorogenic invasiveness and metastatic properties? Could authors comment on this issue and describe if any evidence exists? In a similar manner, do senescent stromal cells gain a fitness advantage over non-senescent functional stroma?
3. Does evidence exist that helps explain melanoma metastasis using stromal senescence mechanisms? Especially in old age, as skin aging might trigger melanoma cells to go and find other locations like lungs that might be more supportive of tumor growth than aged skin. (Fane ME et al, Nature, 2022; Jin J, et al. Signal Transduct Target Ther. 2023)
4. The age-related senescent-like T cells are primarily terminally differentiated and TEMRA cells, effector memory T cells that reacquired naive-like state, and not all T cells truly experience senescence and cell cycle arrest. The authors need to clearly describe such differences when it comes to the description of the aged T cells and their senescence.
5. Throughout the discussion, the authors provided evidence that supports stromal cell senescence and production of SASP modulate different components of the TME and in the majority of the case lead to outcomes in the favor of the tumor. Given the secretory nature of SASP molecules, how in these studies the stromal versus non-stromal SASP effect was distinguished? In other words, how the specificity of stromal SASP was attributed to the measurements in a complex TME?
6. In section 10.2 BOX2, the authors posit that senescence-induction therapy should introduce an artificial burden of senescent cells (how specifically TME is unclear) and target these cells by senolysis. It is difficult to understand why one could want to have more senescent cells in TME, despite so many are already there. Throughout the text, the authors made it clear that different cell types acquire senescence in the TME, and stromal senescence, in particular, represents one of the main culprits.
7. The authors described that the accumulation of senescent cells is primarily driven by the loss/decline of their elimination. It would be interesting to have a short section discussing the age-related reduction in surveillance and elimination of senescent cells by NK and cytotoxic T cells, and if any mechanistic details are available. This can also be helpful in explaining figure 1.
8. There are a few newer reports on the role of stromal senescence in tumor growth and metastasis. They should be included. (PMIDs: 36612020, 36552790, 36038666 , 35650435, 33299638, 33196790)
9. Is there any studies that tested senostatic or senolytic intervention in tumor models? Is there any data available showing the improvement in stromal senescence?
Minor comments.
1. Expand the abbreviations used in the text- DMBA/TPA, HCC, ATM/ AOR, PTEN, OPN, IFITM3, SPINK1, etc.
2. What is Table 38 in Figure 2 legend?
3. A reference(s) is needed for the sentence- "The only certainty is that evSASP ......soluble SASP factors. "
Author Response
Reviewer 2
Major comments:
- Figure 2 needs to modify in a way to simplify the interactions among different pathways that trigger SASP production. There are many interactions described in the legend/ text that do not reflect in the figure, for example, mTOR's ability to activate NFkB directly or in association with mTOR/IL-1R1 signaling did not effectively communicate in the figure. To clearly denote the path of key interacting axes that trigger the final event (NFkB activation, cEBPb binding, or other genetic and epigenetic change), the author can make use of different colors or numbers, so that readers can easily comprehend.
Re: Thank you for the comments. The interaction relationships among mTOR, IL-1A, and NF-κ have been added. In addition, the colors of molecules and pathways have been modified to make them easier to understand.
- What advantage does cancer cell-intrinsic SASP and stromal-derived SASP has on overall tumor progression? Does the senescence of cancer cells result from selection pressure of immunity, therapy, nutritional resources, etc., resulting in the growth and proliferation of selectively fit clones that gain more aggressive tumorogenic invasiveness and metastatic properties? Could authors comment on this issue and describe if any evidence exists? In a similar manner, do senescent stromal cells gain a fitness advantage over non-senescent functional stroma?
Re: Thank you for your comments. Similar issues have been discussed in the already-published report by Milanovic et al [1]. Tumor cells released from senescence acquire a stronger tumor proliferative potential and the stemness of this population appears elevated. This may be because cellular senescence acts as a defense mechanism allowing tumors to respond to harmful stimuli, such as chemotherapy. Related issues have been described in this review. In section 5.2 how cancer cell stemness is linked to cellular senescence has been discussed, which also involves this innovative finding by Milanovic et al [1]. To our knowledge, there are few of literature to discuss the second question, which needs furthermore studies.
- Does evidence exist that helps explain melanoma metastasis using stromal senescence mechanisms? Especially in old age, as skin aging might trigger melanoma cells to go and find other locations like lungs that might be more supportive of tumor growth than aged skin. (Fane ME et al, Nature, 2022; Jin J, et al. Signal Transduct Target Ther. 2023)
Re: Based on the constructive suggestion, we read carefully these two articles on the relationship between the aged microenvironment and melanoma lung metastasis in the elderly and got enlightenment, which has been cited in the revised manuscript. Indeed, changes in the microenvironment of the aged lung, especially alterations in the secretion profile of fibroblasts may be used to explain the development of lung metastases of malignant melanoma. However, we also noted that the relationship of cellular senescence in lung metastases of malignant melanoma is not very clear, although the possibility of involvement exists. Accordingly, section 6 has been modified to complement the discussion on the explanation of distal metastasis of malignant melanoma based on the theory of senescent stroma.
- The age-related senescent-like T cells are primarily terminally differentiated and TEMRA cells, effector memory T cells that reacquired naive-like state, and not all T cells truly experience senescence and cell cycle arrest. The authors need to clearly describe such differences when it comes to the description of the aged T cells and their senescence.
Re: Thanks for your valuable advice. The requested changes have been made to emphasize the fact that not all T cells undergo cellular senescence in section 7.3.
- Throughout the discussion, the authors provided evidence that supports stromal cell senescence and production of SASP modulate different components of the TME and in the majority of the case lead to outcomes in the favor of the tumor. Given the secretory nature of SASP molecules, how in these studies the stromal versus non-stromal SASP effect was distinguished? In other words, how the specificity of stromal SASP was attributed to the measurements in a complex TME?
Re: As mentioned in this review, many studies have confirmed the effect of SASP factors. Some common SASP factors, such as IL-6, IL-8, and MMPs, may be derived from senescent cells. But other types of cells are also capable of secreting these SASP factors. Although references and evidence are cited in our review, some of these studies do not explain the origin of the SASP factors involved. We have described this in detail in the text to ensure that our analysis is objectively detailed.
- In section 10.2 BOX2, the authors posit that senescence-induction therapy should introduce an artificial burden of senescent cells (how specifically TME is unclear) and target these cells by senolysis. It is difficult to understand why one could want to have more senescent cells in TME, despite so many are already there. Throughout the text, the authors made it clear that different cell types acquire senescence in the TME, and stromal senescence, in particular, represents one of the main culprits.
Re: We apologize for the lack of description of the drug-induced subjects in the writing process, which caused some confusion. In this therapy, the target cells of the senescence inducer are cancer cells rather than stromal cells. By inducing senescence in cancer cells, they can be brought to cell cycle arrest. They are then eliminated by senolysis agents. This is a therapeutic strategy that is still in the research phase and was discussed in a previously published review by Wang L et al [2]. Both its short-term and long-term efficacy need to be more explored, and it remains to be proven whether stromal cells will senescence due to off-target effects during this process, and the extent and magnitude of the harm. BOX2 has been further modified to better demonstrate the main process of 'one-two punch' therapy and to reflect as clearly as possible the actual situation of the current study.
- The authors described that the accumulation of senescent cells is primarily driven by the loss/decline of their elimination. It would be interesting to have a short section discussing the age-related reduction in surveillance and elimination of senescent cells by NK and cytotoxic T cells, and if any mechanistic details are available. This can also be helpful in explaining figure 1.
Re: Thank you for the constructive suggestion. The manuscript has presented related context in section 3.3 - "Consistent with this, a recent study found that senescent skin fibroblasts express the non-classical MHC molecule HLA-E, which interacts with the inhibitory receptor NKG2A, expressed by natural killer and highly differentiated CD8+ T cells, to suppress immune surveillance against senescent cells". We would prefer not to add an entirely new section, as we think it would be more appropriate for the section "age-associated accumulation of senescent cells", in addition to space constraints. However, we note that we did not cite the article mentioned in your follow-up suggestion [3], which is useful for understanding the accumulation of senescent stromal cells. Therefore, a discussion of this article in section 3.3 has been included and the paragraphs have been adjusted to make this section more substantial.
- There are a few newer reports on the role of stromal senescence in tumor growth and metastasis. They should be included. (PMIDs: 36612020, 36552790, 36038666 , 35650435, 33299638, 33196790)
Re: These references have been added to make this review more up-to-date.
- Is there any studies that tested senostatic or senolytic intervention in tumor models? Is there any data available showing the improvement in stromal senescence?
Re: Thank you for your comments. Indeed, there are several studies concerning senostatic or senolytic intervention in tumor models [4-6]. However, it is worth noting that these studies focused mainly on tumor cells. Existing studies on the elimination of senescent stromal cells have indeed paid little attention, and therefore the relevant content was relatively omitted in our review. In addition, this review focuses more on the summary of the tumor-promoting effects of senescent stromal cells, while the relevant therapeutic strategies are not the focus. A detailed summary of the studies on senescence-related therapies has been previously conducted [2].
Minor comments:
- Expand the abbreviations used in the text- DMBA/TPA, HCC, ATM/ AOR, PTEN, OPN, IFITM3, SPINK1, etc.
Re: We have carefully reviewed and expanded the related abbreviations in the revised manuscript.
- What is Table 38 in Figure 2 legend?
Re: We are sorry that this error that the words "table 38" should be "p38 MAPK", which has been corrected in the revised manuscript.
- A reference(s) is needed for the sentence- "The only certainty is that evSASP ......soluble SASP factors. "
Re: The references have been added.
References:
[1] Milanovic, M., Fan, D. N. Y., Belenki, D., Däbritz, J. H. M., Zhao, Z., Yu, Y., Dörr, J. R., Dimitrova, L., Lenze, D., Monteiro Barbosa, I. A., Mendoza-Parra, M. A., Kanashova, T., Metzner, M., Pardon, K., Reimann, M., Trumpp, A., Dörken, B., Zuber, J., Gronemeyer, H., Hummel, M., … Schmitt, C. A. (2018). Senescence-associated reprogramming promotes cancer stemness. Nature, 553(7686), 96–100. https://doi.org/10.1038/nature25167
[2] Wang, L., Lankhorst, L., & Bernards, R. (2022). Exploiting senescence for the treatment of cancer. Nature reviews. Cancer, 22(6), 340–355. https://doi.org/10.1038/s41568-022-00450-9
[3] Di Matteo, S., Avanzini, M. A., Pelizzo, G., Calcaterra, V., Croce, S., Spaggiari, G. M., Theuer, C., Zuccotti, G., Moretta, L., Pelosi, A., & Azzarone, B. (2022). Neuroblastoma Tumor-Associated Mesenchymal Stromal Cells Regulate the Cytolytic Functions of NK Cells. Cancers, 15(1), 19. https://doi.org/10.3390/cancers15010019
[4] Johmura, Y., Yamanaka, T., Omori, S., Wang, T. W., Sugiura, Y., Matsumoto, M., Suzuki, N., Kumamoto, S., Yamaguchi, K., Hatakeyama, S., Takami, T., Yamaguchi, R., Shimizu, E., Ikeda, K., Okahashi, N., Mikawa, R., Suematsu, M., Arita, M., Sugimoto, M., Nakayama, K. I., … Nakanishi, M. (2021). Senolysis by glutaminolysis inhibition ameliorates various age-associated disorders. Science (New York, N.Y.), 371(6526), 265–270. https://doi.org/10.1126/science.abb5916
[5] Wang, L., Jin, H., Jochems, F., Wang, S., Lieftink, C., Martinez, I. M., De Conti, G., Edwards, F., de Oliveira, R. L., Schepers, A., Zhou, Y., Zheng, J., Wu, W., Zheng, X., Yuan, S., Ling, J., Jastrzebski, K., Santos Dias, M. D., Song, J. Y., Celie, P. N. H., … Bernards, R. (2022). cFLIP suppression and DR5 activation sensitize senescent cancer cells to senolysis. Nature cancer, 3(11), 1284–1299. https://doi.org/10.1038/s43018-022-00462-2
[6] Wang, C., Vegna, S., Jin, H., Benedict, B., Lieftink, C., Ramirez, C., de Oliveira, R. L., Morris, B., Gadiot, J., Wang, W., du Chatinier, A., Wang, L., Gao, D., Evers, B., Jin, G., Xue, Z., Schepers, A., Jochems, F., Sanchez, A. M., Mainardi, S., … Bernards, R. (2019). Inducing and exploiting vulnerabilities for the treatment of liver cancer. Nature, 574(7777), 268–272. https://doi.org/10.1038/s41586-019-1607-3
Reviewer 3 Report
The manuscript entitled “Senescent stromal cells in the tumor microenvironment: victims or accomplices?” presents the generation of senescent stromal cells in the tumor microenvironment and their specific biological functions. Although the review manuscript is well structured, it should emphasize the main findings. I have some suggestions and questions (please see comments below) since the manuscript could be improved in some parts.
In section 2.2, the following sentences are not clear: Overall, there is abundant evidence that stromal cells within the TME undergo senescence and are highly likely to play an important role in tumorigenesis. However, while these studies confirmed the presence of cellular senescence in the tumor stroma, there is a scarcity of exploration of the types of senescent stromal cells and the role played by cellular senescence in TME.
In Figure 1, mechanisms involved in the formation of the senescent stroma are shown. However, the authors emphasize three factors in the context. Thus, the authors need to modify it by considering the context.
In section 3.2, “The induction from senescent cells in TME,” the authors must discuss whether tumor cell- or senescent tumor cell-induced stromal cell senescence. For example, the authors need to clarify tumor cells or tumor senescence cells in the following sentence: The results of the co-culture of stromal cells with tumor cells also facilitate the understanding of cell crosstalk.
In section 3.3, “Age-associated accumulation of senescent cells,” it is unclear whether immunosenescence induces fibroblast and cancer cell senescence.
The legend of Fig. 2 should be corrected precisely.
The main pathway for inducing SASP in Fig. 2 should be shown according to the contents described as followings: The three canonical pathways that are currently thought to primarily regulate SASP are p38 mitogen-activated protein kinase (p38 MAPK, also known as MAPK14), CCAAT Enhancer Binding Protein Beta (c/EBPβ), and IL-1α.
In section 5.1, cancer stem cells (CSCs) and cancer stemness are unrelated to senescence.
It is unclear whether SASP induces senescence of immune cells or infiltration of immune cells in TME.
In section 8. the authors describe tumor therapy resistance promoted by senescent stroma. However, the main findings have yet to be discussed.
Author Response
Reviewer 3
- In section 2.2, the following sentences are not clear: Overall, there is abundant evidence that stromal cells within the TME undergo senescence and are highly likely to play an important role in tumorigenesis. However, while these studies confirmed the presence of cellular senescence in the tumor stroma, there is a scarcity of exploration of the types of senescent stromal cells and the role played by cellular senescence in TME.
Re: This paragraph has been modified to be clearer and more intuitive.
- In Figure 1, mechanisms involved in the formation of the senescent stroma are shown. However, the authors emphasize three factors in the context. Thus, the authors need to modify it by considering the context.
Re: Thank you for your suggestion. Changes have been made to make the text consistent to reduce the potential confusion caused by the mismatch between the two.
- In section 3.2, “The induction from senescent cells in TME,” the authors must discuss whether tumor cell- or senescent tumor cell-induced stromal cell senescence. For example, the authors need to clarify tumor cells or tumor senescence cells in the following sentence: The results of the co-culture of stromal cells with tumor cells also facilitate the understanding of cell crosstalk.
Re: Indeed, the current studies show that both non-senescent and senescent tumor cells promote stromal cell senescence under co-culture conditions. Additions have been made to highlight the differences between senescent and non-senescent tumor cells in the induction of stromal cell senescence and cell crosstalk.
- In section 3.3, “Age-associated accumulation of senescent cells,” it is unclear whether immunosenescence induces fibroblast and cancer cell senescence.
Re: This section has been fine-tuned to make it more objective and appropriate to the current situation.
- The legend of Fig. 2 should be corrected precisely.
Re: The legend of Fig. 2 has been revised precisely.
- The main pathway for inducing SASP in Fig. 2 should be shown according to the contents described as followings: The three canonical pathways that are currently thought to primarily regulate SASP are p38 mitogen-activated protein kinase (p38 MAPK, also known as MAPK14), CCAAT Enhancer Binding Protein Beta (c/EBPβ), and IL-1α.
Re: Thank you for your valuable advice. These three signaling pathways do play a key role in regulating SASP, but this is only the tip of the iceberg. Therefore, this figure were drawn to show the molecular processes involved as comprehensively as possible. And Fig. 2 has been further modified with your suggestion to show the SASP regulatory network in a comprehensive way.
- In section 5.1, cancer stem cells (CSCs) and cancer stemness are unrelated to senescence.
Re: Senescence and cancer stem cells seem to be linked, as pointed out in previous studies [1, 2]. However, these studies mainly focused on the molecular features shared between cancer stem cells and senescent cancer cells, while the connection between cellular senescence, stemness, and stromal cells is not very clear. There may have been some misunderstanding because our article did not emphasize this aspect, so section 5.2 has been revised to ensure that this section is more accessible and objective.
- It is unclear whether SASP induces senescence of immune cells or infiltration of immune cells in TME.
Re: Indeed, whether SASP factors, especially stromal-derived ones, modulate immune cell infiltration remains a direction still under investigation. However, some of these important SASP factors, such as IL-6 and IL-8, have been shown to have an inducing effect on the senescence of immune cells and the infiltration of immune cells in TME. As it stands, a question worth discussing is where these soluble substances come from in Vivo. If they are secreted by senescent stromal cells, then it can be argued that SASP factors induce the senescence of immune cells and the infiltration of immune cells in TME. Although there is still controversy on the current study, we, therefore, wrote this part in order to make the article more comprehensive. With the comments provided, it has been revised to explain more clearly the current status of the study in the first paragraph of section 7.
- In section 8. the authors describe tumor therapy resistance promoted by senescent stroma. However, the main findings have yet to be discussed.
Re: The mechanisms of tumor therapy resistance induced by senescent stroma are intricate and related to several mechanisms presented in previous sections, and therefore it is mentioned that “The synergistic effects of senescent stroma on neoplasm growth are convoluted, but in general, there are two aspects, direct tumor progression promotion (EMT, CSC generation, angiogenesis, ECM remodeling, distant metastasis, etc.) and indirect influence on tumor immunity (inflammation promotion, immunosuppressive cell recruitment, T cell senescence induction, etc.), the detailed evidence of which has been initially discussed above”. The discussion of molecular mechanisms in this section is indeed somewhat lacking, but we think it is appropriate. Because it is more like a summary showing that the tumor-promoting effect of the senescent stroma ultimately causes tumor therapy resistance. However, your comments did enlighten us, and they have been adapted appropriately.
References:
[1] Milanovic, M., Fan, D. N. Y., Belenki, D., Däbritz, J. H. M., Zhao, Z., Yu, Y., Dörr, J. R., Dimitrova, L., Lenze, D., Monteiro Barbosa, I. A., Mendoza-Parra, M. A., Kanashova, T., Metzner, M., Pardon, K., Reimann, M., Trumpp, A., Dörken, B., Zuber, J., Gronemeyer, H., Hummel, M., … Schmitt, C. A. (2018). Senescence-associated reprogramming promotes cancer stemness. Nature, 553(7686), 96–100. https://doi.org/10.1038/nature25167
[2] Gorgoulis, V., Adams, P. D., Alimonti, A., Bennett, D. C., Bischof, O., Bishop, C., Campisi, J., Collado, M., Evangelou, K., Ferbeyre, G., Gil, J., Hara, E., Krizhanovsky, V., Jurk, D., Maier, A. B., Narita, M., Niedernhofer, L., Passos, J. F., Robbins, P. D., Schmitt, C. A., … Demaria, M. (2019). Cellular Senescence: Defining a Path Forward. Cell, 179(4), 813–827. https://doi.org/10.1016/j.cell.2019.10.005